# Targeting the Gut in Obesity: Signals from the Inner Surface

**DOI:** 10.3390/metabo12010039

**Published:** 2022-01-05

**Authors:** Natalia Petersen, Thomas U. Greiner, Lola Torz, Angie Bookout, Marina Kjærgaard Gerstenberg, Carlos M. Castorena, Rune Ehrenreich Kuhre

**Affiliations:** 1Global Obesity and Liver Disease Research, Global Drug Discovery, Novo Nordisk A/S, Novo Park 1, 2670 Måløv, Denmark; ltpn@novonordisk.com (L.T.); mikg@novonordisk.com (M.K.G.); ruku@novonordisk.com (R.E.K.); 2The Wallenberg Laboratory and Sahlgrenska Center for Cardiovascular and Metabolic Research, Department of Molecular and Clinical Medicine, Institute of Medicine, University of Gothenburg, 405 30 Gothenburg, Sweden; Thomas.Greiner@wlab.gu.se; 3Department of Veterinary and Animal Science, Faculty of Health and Medical Sciences, University of Copenhagen, 2200 Copenhagen, Denmark; 4Global Obesity and Liver Disease Research, Global Drug Discovery, Novo Nordisk Research Center, Seattle, WA 98109, USA; gabt@novonordisk.com (A.B.); ccto@novonordisk.com (C.M.C.)

**Keywords:** appetite regulation, enteroendocrine cells, enterokines, gut microbiota, nutrient metabolism, therapeutic potential, intestinal remodelling

## Abstract

Obesity is caused by prolonged energy surplus. Current anti-obesity medications are mostly centralized around the energy input part of the energy balance equation by increasing satiety and reducing appetite. Our gastrointestinal tract is a key organ for regulation of food intake and supplies a tremendous number of circulating signals that modulate the activity of appetite-regulating areas of the brain by either direct interaction or through the vagus nerve. Intestinally derived messengers are manifold and include absorbed nutrients, microbial metabolites, gut hormones and other enterokines, collectively comprising a fine-tuned signalling system to the brain. After a meal, nutrients directly interact with appetite-inhibiting areas of the brain and induce satiety. However, overall feeding behaviour also depends on secretion of gut hormones produced by highly specialized and sensitive enteroendocrine cells. Moreover, circulating microbial metabolites and their interactions with enteroendocrine cells further contribute to the regulation of feeding patterns. Current therapies exploiting the appetite-regulating properties of the gut are based on chemically modified versions of the gut hormone, glucagon-like peptide-1 (GLP-1) or on inhibitors of the primary GLP-1 inactivating enzyme, dipeptidyl peptidase-4 (DPP-4). The effectiveness of these approaches shows that that the gut is a promising target for therapeutic interventions to achieve significant weigh loss. We believe that increasing understanding of the functionality of the intestinal epithelium and new delivery systems will help develop selective and safe gut-based therapeutic strategies for improved obesity treatment in the future. Here, we provide an overview of the major homeostatic appetite-regulating signals generated by the intestinal epithelial cells and how these signals may be harnessed to treat obesity by pharmacological means.

## 1. Introduction

The digestive system is one of the oldest systems in the evolution of mammals, carrying out the remarkable task of providing a defence against the outside world while facilitating the passage of essential macronutrients and vitamins. While supplying nutrients to the whole body is the main function of the intestine, the mediators produced in this process control food intake. Intestinally derived signals act on receptors on vagus nerves and neurons in appetite-controlling areas of the brain. In obesity, satiety signalling is inhibited, resulting in excessive food intake. Although a vast number of studies have been carried out to dissect the mechanisms that prevent the normal satiety in individuals living with obesity, the identity of the driving factor(s) is lacking. Most studies on obesity pathogenesis suggest the main cause is central dysregulation of feeding behaviour and lack of hunger suppression in the brain. Impaired gut hormone secretion and dysbiosis are often associated with obesity, but it is unclear whether this perturbed signalling has a causative role or whether its long-term perturbed signalling persists after weight loss and/or weight regain. Nevertheless, the signals derived from the intestine in the process of nutrient processing and uptake can be extremely powerful to supress the appetite. For example, the recent success of the GLP-1-based analogues for weight loss provides clear evidence of the potential for utilizing the intestinally-derived signalling pathway for obesity treatment. However, the intestine itself as a drug target can provide many more ways of modulating the energy intake.

Nutrient signalling is the primary signal initiating satiety at several levels and each one of them offers an opportunity to influence the satiety circuits—such as modulating the absorption rate or gut hormone secretion. Keeping in mind that the intestine is the largest inner organ in the body and contributes with 20–35% of whole-body energy expenditure [1], the diet-induced energy expenditure component is important for energy homeostasis. The intestinal epithelial lining is heavily vascularized and equipped with an underlying network of lymphatic capillaries which not only allow nutrient absorption but also interaction with immune cells. Recent studies show that these interactions are tightly connected to the satiety signalling. In addition, metabolites from symbiotic microorganisms, contributing to digestion, form signalling pathways to various systems of the human body, such as liver, immune system, and brain, helping to maintain energy homeostasis. As impaired regulation of food intake is the main driving force in obesity, focusing on origins of the natural satiety signalling and developing new ways of boosting it may provide us with better and safer ways of controlling the appetite.

In this review, we will describe the satiety feedback signals generated by the intestine during digestion and how these pathways may be harnessed to control the food intake and thereby produce weight loss.

## 2. Intestinal Epithelium and Signalling Properties of Nutrients

The intestinal epithelial cell layer is a highly adaptive tissue that is almost completely renewed every week. It consists of more than 30 cell types, all originating from one type of stem cell expressing leucine-rich repeat containing G protein-coupled receptor 5 (Lgr5+). The process of intestinal cell differentiation and their renewal cycle has been described earlier in several excellent reviews [2,3]. The ratio of absorptive and secretory cells differentiating from Lgr5+ cell progeny, transit-amplifying cells (Figure 1), is regulated by Notch signalling. As shown in mouse studies, the cell plasticity in the proliferative compartment can have an important role for ratios of specific cells types in the intestinal epithelium. This has a direct effect on both absorptive and signalling properties [4,5] and thereby might increase the risk of obesity-related complications [6,7]. Importantly, obesity and high calorie food intake is associated with enhanced Wnt signalling in the stem cell population [8], which results in increased proliferation. This, in turn, favours enterocyte differentiation over secretory cells [9]. Although this process has not been sufficiently studied, such remodeling may be associated with increased capacity to absorb nutrients. Enteroendocrine cells (EECs) constitute only 1–2% of all intestinal epithelial cells. Despite their small numbers they produce more than 15 different hormones that collectively play a key role in regulation of food intake and overall metabolism. Other intestinal secretory cells are exocrine cells, such as mucus producing goblet cells, Paneth cells that produce growth factors and immune cell-like products, as well as tuft cells that regulate epithelial sentinels by sensing the luminal content, secreting immune factors and by interacting with immune cells [10].

During digestion, complex macronutrients are broken down by enzymes from saliva and exocrine pancreatic secretions by gastric acid and by bile supplied from the liver. Absorption of nutrients is carried out by enterocytes, the most abundant cell type in the intestinal epithelium, and occurs particularly in the upper part of the small intestine. Enterocytes also actively secrete ions into the gut lumen to regulate associated water influx/outflux during the active phase of absorption. Nutrients, which are breakdown products of carbohydrates, lipids and proteins, are absorbed through the enterocyte’s surface. Some of the larger molecules, such as chylomicrons, are transported into the lacteals-lymphatic capillary spaces. These feedback signals are released into the circulation to produce a feeling of satiety in the brain through specific receptors on hypothalamic and hindbrain neurons [for review [11]].

This sensory system is evolved to prepare the body for digestion and deposition of different types of macronutrients. The appetite-regulating effects mediated by nutrients rely to a large degree on highly specialized G-protein coupled receptors (GPCRs or GPRs), which provide instant and precisely regulated responses in the brain and other organs (Figure 1). After the absorption of nutrients, non-digested food components, such as fibres and products of digestive processes such as bile acids, are further metabolised by intestinal microbiota in the terminal part of the small intestine and in the large intestine. Many of these products also become local and circulating signalling molecules and steer the host metabolism through modulation of gut hormone secretion or by direct interaction with other organs (as described further in *Microbial metabolites*).

Diet modifications to reduce energy intake remains the main way of managing obesity [12,13,14]. However, following an initial rapid weight loss and a plateauing of weight loss typically occurring at one year, a gradual regain is normally experienced by most individuals [15,16]. Therefore, enhancing the satiety-inducing properties of the gut may prove a better long-term strategy for weight loss by helping to maintain reduced caloric intake, and potential targets are manifold. For example, some of the dietary components, such as fibres or medium chain triglycerides prolong or/and accelerate the satiety after a meal. While most of the saturated fats are packed as chylomicrons and then absorbed into the lacteals, medium chain fatty acids can be directly absorbed into the blood stream and invoke satiety signalling faster. Indeed, the satiety inducing effects of medium chain triglycerides are currently being exploited as anti-obesity treatment in the form of dietary supplements.

Fat sensing by itself has satiating power even before the nutrients reach the bloodstream. Different types of fats entrain FA transporters (e.g., CD36) and GPCR (e.g., FFAR1 (GPR40), FFAR2, FFAR3, FFAR4 (GPR120), and GPR84); and Delayed Rectifying K+ (DRK) channels (for review [17]). Activation of these receptors modulates secretion of gut hormones (addressed in *Enteroendocrine cells*) and thereby enables satiety signalling through the gut-brain axis. Apart from that, some of the lipids have a direct effect on satiety and energy homeostasis centres in the hypothalamus, such as polyunsaturated fatty acids through GPR120 (mostly expressed on microglia) and GPR40 [13]. In fact, a study on mice showed that reduced expression of these two receptors increases energy efficiency [18], whereas central GPR120 agonism acutely inhibits food intake [19]. These two receptors are also expressed in the gut and their activation increases glucose-dependent insulinotropic peptide (GIP) and glucagon-like peptide-1 (GLP-1) secretion [20,21,22].

Products of protein digestion-peptides, oligopeptides and amino acids-are sensed by other receptors, such as the calcium-sensitive-receptor (CaSR) and the umami receptor (T1R1–T1R3). Importantly, the gut hormone release activated by these receptors has a very strong inhibitory effect on the appetite, which makes proteins very powerful satiety signals [23].

Overall, the three types of macronutrients—proteins, lipids and carbohydrates—have a major role in satiety signalling by generating negative feedback signals in various parts of the intestine [24]. Early studies, in normal weight people, showed that isocaloric intake of sugar and fat had similar inhibitory effects on further food intake. However, in people living with obesity, carbohydrate rich foods have been shown to suppress food intake to a larger extent than high fat foods (when adjusted for calorie content [25]. On the contrary, another study showed that when an isocaloric stimulus of either fat or sugar was injected directly into the duodenum, fat inhibited food intake more than sugar (~22 vs. ~15%), and in this case there was no difference between normal-weight individuals and individuals living with obesity [26]. Moreover, site-specific intestinal detection of macronutrients differentially inhibits the AGRP neurons in the hypothalamic arcuate nucleus, as described in mouse studies. Thus, vagal gut-brain signalling is required for lipid signalling, while spinal gut-brain signalling relays the presence of intestinal glucose [27]. Lipids may be better at signalling satiety, but they are much more calorie dense compared to carbohydrates. On the other hand, carbohydrates can drive hedonic impulses for food intake–in the limbic system–which may have the leading role in over-consumption of food beyond homeostatic needs [28]. These are a few examples of the crosstalk between the gut and the brain initiated by macronutrients and how they may promote the development of obesity. Thus, in the context of factors driving obesity, the relative contribution of fat and carbohydrates is controversial. The lack of consensus in studies investigating the satiety properties of various macronutrients results from differing macronutrient sources and ratios of macronutrients in tested meals, and therefore more standardized studies should be performed.

While intestinal absorption does not seem to be a changed in calorie restriction or weight gain [29], the signalling along these lipid and carbohydrate pathways is often altered in metabolic diseases associated with prolonged energy surplus. Adaptation of intestinal epithelial cells to augmented caloric overload increases villus length and thereby the capacity for the nutrient absorption rate [8]. The structural remodelling of enterocytes in obesity also increases the absorptive capacity of enterocytes themselves for high calorie nutrients [28] as well as their capacity to store more fat in the cytoplasm. Interestingly, a study on high fat diet-fed mice showed that increasing the fatty acid oxidation in enterocytes reduces the accumulation of visceral fat [30]. Taking into the account the size of the intestine and energy burning capacity, targeting the breakdown of lipids already in the intestinal epithelium could be a therapeutic strategy against obesity, as it will decrease lipid absorption into the blood and thus protect against nutrient overload.

Orlistat, a saturated derivative of lipostatin and potent natural lipase inhibitor, is currently used for weight loss by reducing lipid absorption from the lumen [31]. Similar approaches have been developed to target glucose absorption in obesity and diabetes. Glucose is mainly absorbed by sodium-glucose transprorter-1 (SGLT1). Phloridzin, a naturally occurring dihydrochalcone from root bark of apple trees, blocks SGLT1 in the gut and SGLT2 in the proximal renal tubules thereby reducing both absorption of glucose and reabsorption in the kidney [32]. However, the use of this drug was discontinued due to side-effects, mostly osmotic diarrhea. Currently, another SGLT1/2 inhibitor is being evaluated for treatment of type-1-diabetes (Sotagliflozin (LX4211), and studies in rodents with this compound have shown a beneficial effect for weight loss [33].

Novel drug delivery strategies with a selective action in the intestine may be particularly effective and safe, as the lumen is naturally separated from the systemic blood flow and because the cell renewal allows for making quick adjustments to dosing regimens. As an example, intestinal hydrogel implants containing a biologically active agent, epigallocatechin gallate, showed to be effective for preventing fat absorption in obese mice.

The feeling of satiety is also regulated by mechanoreceptors in the stomach and in the intestine, and can be enhanced by dietary fibre intake, which delays the emptying of the stomach and the absorption of nutrients. In addition to its effect on appetite, diets rich in fibre content have been shown to provide numerous health benefits, including improved glucose tolerance and a reduction in cholesterol levels. However, the effect of fibre-rich foods on weight loss in clinical trials is modest [34].

To conclude this section, while pharmacological modulation of direct nutrient sensing or absorption seems a viable option to reduce caloric intake, there are no approved drugs that target nutrient sensing mechanisms, and the only existing drug that targets absorption is Orlistat, which only provides a moderate weight loss and often is accompanied by unpleasant side effects. However, as morbid obesity must be managed with a multi-therapy approach, combining drugs with lifestyle interventions may allow reducing the effective dose to minimize the side effects.

## 3. Enteroendocrine Cells

When it comes to the most powerful satiety signals, EECs are the main cells informing the appetite centres of nutritional status in the gut. They descend from the same Lgr5+ stem cells as all other intestinal epithelial cells, and during their differentiation acquire a sophisticated sensing machinery that allows for a tight regulation of hormone release. Despite the low relative abundance (1–2% of intestinal cells) EECs represent the largest endocrine organ in the body [35] (Figure 2).

The majority of EECs are serotonin, also called 5-hydroxytryptamine (5-HT)-secreting cells, which regulate motility and intestinal cell differentiation. Serotonin cells have very few nutrient receptors, but express receptors for GLP-1 and for some microbial metabolites [36]. Autocrine and paracrine signaling of intestinal serotonin is investigated for correction of inflammation and intestinal motility as well and weight management [37]. Long-acting compounds targeting serotonin signaling in the brain, such as sibutramine (a 5-HT uptake inhibitor) and Locaserin (a selective 5-HT2cR inhibitor) [38], were used for obesity treatments, but were withdrawn by the FDA. The role of intestinally-produced serotonin in energy homeostasis is not clear. Dysregulated serotonin signaling is observed in morbid obesity [39]. On the other hand, a study in mice showed that the reduction of peripheral serotonin levels increases thermogenesis, so it may have a negative effect on the energy balance in obesity [40]. Investigations to determine if this result translates to humans would be insightful.

Although enteroendocrine hormones were originally described to be produced and secreted by distinct EEC-types, newer research has shown that the different EECs subtypes often co-express several anorectic hormones with similar, but not completely overlapping effects. The most important in this regard are cholecystokinin (CCK), GIP, GLP-1, neurotensin, secretin, and peptide YY (PYY), which all are released during and following meal intake to reduce appetite and improve nutrient utilization and disposition. A number of thorough reviews describing the sensory machinery in EECs can be found elsewhere for nutrient and metabolite signalling [41,42]. Many intestinally produced hormones have direct targets in the central nervous system, but they also act on the brain indirectly through the vagus nerve and the enteric nervous system, commonly referred to as the gut-brain-axis. Modern therapies utilizing gut hormone signalling are mostly based on the anorexigenic effects. It has been shown that caloric overload is associated with blunted secretion of GLP-1, pancreatic polypeptide [43] and CCK [44]. Although the causative role of these blunted secretions for metabolic diseases is unclear, increasing signalling of these hormones by pharmacological means has been repeatedly shown to induce numerous beneficial metabolic effects. While synthetic analogues of appetite-inhibiting hormones are effective means for weight loss and blood sugar control, enhancing gut hormone secretion may also be a suitable (if not superior) alternative. In support of this notion, studies of individuals after a gastric bypass operation (sleeve gastronomy or Roux-en-Y gastric bypass) show that increased endogenous secretion of gut hormones after the operation is key to the beneficial effects on weight loss [45]. GLP-1 is arguably the gut hormone that shows the largest relative increase, with post operation secretion up to 30-fold [46] over normal. However, CCK, PYY and neurotensin secretion also increases and acts in concert with increased GLP-1 secretion to inhibit food intake. Indeed, GLP-1 and PYY have been shown to contribute equally to postoperative reduction in food intake [45]. Compared to a monotherapy (e.g., with a GLP-1R agonist), simultaneously increased secretion of several hormones could lead to superior food intake inhibition and corresponding weight loss. Since the overlap of expression profiles of nutrient sensors (e.g., GPR120) between different types of ECC is substantial [47], it is likely that pharmacological targeting of gut hormone secretion could mimic the amplifying effect of gastric bypass on gut hormone release. Another important difference between treatments based on administration of modified gut hormones and targeting mobilization of the body’s own gut hormones stores is pulsatility. Under normal physiological circumstances, anorexigenic gut hormones are secreted in a pulsatile fashion with low secretion at fasting and elevated secretion in response to meal intake. Plasma concentrations of most gut hormones usually peak between 15 and 30 min after meal initiation and concentrations are returned to pre-meal levels within 2–3 h (except for ghrelin that shows the opposite dynamic) [46]. As prolonged high concentrations of gut hormones in plasma may inactivate the receptors, the pulsatile secretion pattern ensures that the respective receptors for the gut hormones get some relief in between meals. For example, continuous infusion of native GLP-1 (7-36 amide) over the course of 8 h leads to tachyphylaxis and reduced inhibition of gastric emptying [48]. However, treatment options based on increasing gut hormone secretion has yet to be approved for obesity treatment, so it remains to be established whether the suspected pulsatile way of action will lead to improved efficacy on weight loss.

Of the appetite inhibiting hormones mentioned above, PYY and GLP-1 are the two hormones that have received the most attention in terms of obesity treatment. GLP-1 based analogues have been used as the standard care treatment of obesity for more than five years. Other appetite-modulating hormones are continuing to be evaluated as stand-alone treatments, as in the case of CCK analogue [49] or in combination with GLP-1 and GIP (e.g., the combined GIP and GLP-1 receptor co-agonist Tirzepatide from Eli Lilly).

Increasing the secretion of GLP-1 by mimicking nutrient challenge with small molecule GPCR agonists has been attempted for several years. The search for incretin secretagogues resulted in development of several pharmacological agents [50,51] targeting the EECs, but so far none of the drugs has been approved for obesity treatment.

Lipid metabolites endocannabinoids act as satiety messengers as GLP-1 secretagogues. Among those, oleoylethanolamide, anandamide and palmitoylethanolamide, produced by intestinal cells, increase GLP-1 by activating GPR119 [51]. In fact, several GPR119 agonists made it to clinical trials as antidiabetic agents, but the efficacy for inducing GLP-1 secretion and for restoring blood glucose control was limited [52].

In addition to increasing GLP-1 and other gut hormone secretion by mimicking the action of nutrients through their receptors, the production of GLP-1 can be modified by the targeted modulation of intestinal cell differentiation. Thus, the inhibition of Notch signalling, favouring differentiation of enteroendocrine cells, can increase the number of GLP-1 producing cells with 3-fold, as well as other secretory cells types [53] (also shown in Figure 3).

Such modulation increased plasma GLP-1 levels and restored glucose tolerance in obese mice, but follow-up studies and food intake measurements were not conducted. However, this modulation interferes with normal epithelial cell renewal and non-endocrine functions of the intestine [53]. A directed targeting of enteroendocrine precursors in the intestinal epithelium would be a safer approach, but may not reach the clinical efficacy as the percentage of these precursor cells is quite low. For example, a treatment with a TGR5 agonist or short chain fatty acid doubles the number of L-cells in the distal small intestine, but it is not known whether this modest modulation is sufficient to restore the satiety signalling in people living with obesity [54]. To conclude, further studies are needed to establish the viability of this unconventional strategy as an anti-obesity therapy.

Although L-cells (Figure 2) are in many ways activated by direct contact with nutrients, the highest density of L-cells are found in the colon, where nutrient luminal content is least abundant. The extent to which colonic L-cells are involved in regulation of meal termination is being debated, but it is clear that L-cells in the colon have a different transcriptional profile compared to ileal L-cells [55], suggesting different functions of L-cells in different parts of the intestine [56]. For example, colonic, but not ileal L-cells produce an orexigenic gut hormone INSL5 [57]. Furthermore, studies on germ free mice showed increased levels of both INSL5 and colonic GLP-1. Germ free mice have very low levels of SCFAs due to the lack of microbes producing them, resulting in the poor energy state of colonocytes. Therefore, we previously suggested that colonic GLP-1 and INSL5 are both part of a response to the famine state when colonic GLP-1 slows intestinal transit and INSL5 increases hepatic glucose production [58]. This is supported by observations in anorectic patients that both have increased GLP-1 levels and slower gastrointestinal transit [59,60], and that INSL5 has shown orexigenic properties during conditions of energy deprivation [57,58].

While the majority of gut hormones are appetite-inhibiting and are secreted in response to a meal, the gastrointestinal tract also generates hunger signals, such as ghrelin, produced by neuroendocrine X/A-like cells, located in the stomach [61] and to a lesser degree in the small intestine. In fact, ghrelin secretion increases between meals and it acts as a driver to initiate meals [62]. This effect is driven by activation of orexigenic neuropeptide-Y (NPY)-secreting neurons of the arcuate nucleus in the hypothalamus, which both stimulate meal intake on their own and by subsequent inhibition of anorexigenic proopiomelanocortin (POMC) neurons [63,64,65].

Furthermore, in the ventromedial nucleus, ghrelin signalling reduces fatty acid synthase levels, and serves as a low energy signal [65,66,67]. In the pituitary, ghrelin induces growth hormone secretion, which plays a crucial part in maintaining glucose levels during calorie restriction in mice [68,69]. Ghrelin also has paracrine effects in the gastrointestinal tract, increasing gastric motility and secretion of gastric acid [70,71]. It is not clear whether ghrelin affects the secretion of incretins in a physiological setting. Several in vitro studies reported a potentiating effect of ghrelin or the ghrelin receptor (GPR39) agonist on GLP-1 secretion in human and mouse L-cells, which appeared to be a potential anti-diabetes strategy [72,73]. However, ghrelin infusion to humans or administration of ghrelin to isolated perfused mouse intestine affected neither plasma GLP-1 concentrations nor GLP-1 output from the intestine [74]. This suggests that while activation of GPR39 may modify GLP-1 secretion to some extent, it may not be sufficient as a therapeutic treatment for obesity.

Ghrelin signalling inhibition, on the other hand, seems a plausible option for therapeutic targeting. Ghrelin exists as acyl-ghrelin (active ghrelin) and des-acyl ghrelin, and only the active ghrelin binds and activates the ghrelin receptor, GHSR [75]. The acylating enzyme, ghrelin-*O*-acyltransferase (GOAT), was recently proposed as a therapeutic target for obesity and type 2 diabetes [76] based on the assumption that it would reduce the production of active ghrelin and thereby decrease meal frequency. Indeed, several studies have shown that GOAT inhibitors enable pharmacological control of ghrelin signalling (reviewed in [77]) and clinical trials are underway using small molecule GOAT inhibitors for suppression of overeating (Clinical trial NCT03641417) and another form of addiction-alcohol abuse (Clinical trial NCT03896516).

In conclusion, the gut hormone based therapies have proven effective for reducing body weight and their efficacy appears to be improved by entraining several gut hormones. However, the precise combination and dosing regimens is the key and needs to be determined to provide clinical success.

## 4. Microbial Metabolites

The microorganisms that colonize our gut, collectively referred to as the gut microbiota, are composed of roughly 39 trillion cells, comparable to the number of human cells in our body [78]. The combined genomes of the microbiota consist of more than 10 million genes, many of which encode enzymes that interact with the nutritional content of the gut to produce an extensive number of metabolically active compounds. These substances affect various host signalling pathways, including those controlling dietary intake and host metabolism. The majority of microbial species inhabit the distal gut, where, in the process of their livelihood, they produce metabolites beneficial for the host, such as vitamins, essential amino acids, short chain fatty acids (SCFAs), and bile acids (reviewed in Singh RK 2017).

SCFAs, acetate propionate and butyrate, are a major energy source for colonocytes, but they can also act as signalling molecules. SCFAs can bind to GPR41 and GPR43 (FFAR3 and FFAR1) to stimulate secretion of GLP-1 from L-cells [79,80]. They also act as histone deacetylase (HDAC) inhibitors [81], key regulators of metabolism that increase thermogenesis in fat tissue. It has also been shown that propionate and butyrate can stimulate production and release of the anorexigenic hormone PYY from human EECs [82].

Even more potent stimulators of GLP-1 release from L-cells are the microbially produced bile acids litocholic acid (LCA) and deoxycholic acid (DCA). Bile acids are produced and secreted by the liver in their so-called primary form, particularly in response to fat-rich meals. This process is also influenced by CCK secretion, which acts to contract the gallbladder to increase bile acid secretion [83]. Primary bile acids are mostly conjugated to either taurine or glycine, which increases their solubility by turning them into anions (bile salts) [84,85,86]. During passage down the intestine, these bile acids are reabsorbed by the secondary active transporter ASBT/IBAT (Apical Sodium-Dependent Bile acid Transporter/Ileal Bile Acid Transporter). However, a substantial amount of bile acids reaches the lower part of the intestine, where microbes de-conjugate and dehydroxylate those to form secondary bile acids, such as LCA (derivative of the primary bile acid chenodeoxycholic acid) and DCA (derivative of the primary bile acid cholic acid). These secondary bile acids are powerful metabolic mediators acting primarily on the two bile acid/salt sensitive receptors, TGR5 and FXR [87,88,89]. The deconjugation dramatically reduces solubility but has little or no effect on affinity and activation of the bile acid receptors. In contrast, dehydroxylaton (or conversion to secondary bile acids) has a considerable impact on both receptor affinity and efficacy. Therefore, microbially-produced lithocholic acid and deoxycholic acids are superior to primary bile acids (chenodeoxycholic acid and cholic acid) in stimulating GLP-1 secretion through TGR5 [87,88,89,90]. Furthermore, bile acids can also negatively regulate GLP-1 production through the nuclear receptor FXR in L-cells. Inhibition of FXR via remodelling of the microbiome in the intestine targeting hepatic lipid metabolism could possibly also be a target for anti-obesity drug [90].

The microbial community can also ferment different types of amino acids, resulting in the production of a vast array of bioactive compounds that can affect host physiology [91,92]. For example, the tryptophan-derived compound indole has shown to affect GLP-1 secretion in L-cells [93] and the histidine-derived metabolite imidazole propionate has shown to have a negative effect on glucose metabolism as well as insulin signalling [94]. Lean and obese individuals have different compositions of distal gut microbiota and their metabolites [91]. This likely reflects nutrient overload as this is associated with induction of rapid changes in the bacterial composition of the human gut microbiota caused by nutrient-dense diets [92]. Furthermore, a study in ob/ob mice showed that their faecal microbiota has an increased capacity to harvest energy including increased absorption of monosaccharides from the gut [95].

In humans, certain traits have been reported to associate with the obese phenotype. Our body relies on bacterial enzymes to break down fibre-rich foods to be absorbed, and several types of bacteria are more efficient in doing so, which may increase the availability of nutrients. For example, two types of organisms were identified to be specific to people living with obesity [96]: the hydrogen-producing bacteria Prevotellaceae, and hydrogen-consuming methanogens Archaea. These bacteria boost the production of fatty acids, which, over time, may promote positive energy balance. On the other hand, Akkermansia sp. and lactate metabolizers were increased in patients following gastric bypass surgery [97]. While the idea of microbiome contributing to human body homeostasis is undoubtedly interesting, the question is how much of caloric absorption can be “vented” by tailoring the microbiome? This is because microbiota is mostly present in the large intestine, while absorption of nutrients, particularly from high calory foods, occurs mostly in the upper small intestine. In addition, the excreted calories can represent less than 10 percent of total calorie intake (which also includes the un-digestible components) [98], so the modifiable fraction is even less.

Faecal transplants from healthy donors with a mix of bacterial strains have shown positive metabolic effects (reviewed in [99]), including increased insulin sensitivity in obesity [100,101], while another study report no significant metabolic improvements [102]. High calorie foods seem to increase relative abundance of Firmicutes species and reduce abundance of Bacteroidetes species in humans [103], however, the use of Firmicute/Bacteroidetes ratio as an obesity hallmark is still being disputed [104]. Another species negatively associated with obesity is Akkermansia muciniphila [105], and dietary supplementation with this microbe not only has beneficial effects on the liver function and inflammatory profile of the treated individuals, but also results in some weight loss [106]. An anti-obesity effect was also shown for endocannabinoid-producing bacteria [107]. Normally, stimulation of endocannabinoid receptors in the brain results in increased food intake. However, the effect is opposite for intestinal signalling of endocannabinoids, such as N-acylphosphatidylethanolamines and their derivatives N-acylethanolamines, which are formed locally in the intestine by microbial species. A two-week administration of genetically engineered *E. coli* to produce N-acylphosphatidylethanolamines decreased the high fat diet-induced weight gain and food intake in mice [107]. Interestingly, this effect was not present in mice on regular chow.

The effect of probiotic strains in obesity in human trials has shown some effects although the effect size generally has been moderate. Specifically, several studies using different strains of *Lactobacillus*, *Streptococcus* and *Bifidobacterium* have shown small effects on weight loss (<1 kg) [108,109], while others showed no difference in body weight after the treatment [110]. Furthermore, supplementation with pasteurized *Akkermansia muciniphila* enhanced weight loss more than live bacteria [106]. Although the described effects on weight loss seem rather subtle, these studies are encouraging as they demonstrated weight losses resulting from relatively short treatments and often by use of only a single, well-studied bacterial strain. As such, these studies can be viewed as encouraging proof-of-concept studies to inspire further studies with longer treatments, use of new bacterial strains and their combinations.

## 5. Enterokines and Gut-Liver Axis

Non-endocrine enterocytes also produce factors with endocrine action, although these enterokines are not packed in large dense core granules, and their secretion is constitutive and do not require tightly regulated sensing machinery. Fibroblast growth factor (FGF) 15/19 (FGF19 in humans, FGF15 in rodents) is one of those enterokines. It is produced mostly in mature enterocytes in the ileum, and to a lesser extent in the jejunum and duodenum [111,112]. Many FGFs are growth factors with important functions for structural integrity of tissues, particularly of its mesenchymal component. However, some FGFs have low affinity for heparin sulphate glycosaminoglycans and thus are secreted into the circulation and as act as signalling molecules. These so called “endocrine” FGFs diffuse into the circulation and act on other organs in the body. This endocrine FGF signalling requires a co-receptor, with FGF15/19 specificity conferred via the FGFR1/beta klotho (KLB) receptor/co-receptor complex [113]. FGF15/19 production is induced when bile acids bind and activate the nuclear receptor FXR in the intestine. FGF15/19 functions as a signalling molecule by suppressing bile acid synthesis when bile acid levels are high in the intestinal mucosa, thus affecting bile receptor FXR signalling [114,115,116,117]. This bile acid-FGF15/19 signalling represses hepatic lipogenesis (which is abnormally high in obesity) [118] and inhibits gluconeogenesis in the liver [116,117,118,119]. When administered in supraphysiological levels or as pharmacological analogues, FGF15/19 also affects the adipose tissue and the brain, presumably through actions on FGFR1/KLB complexes [112,113,120,121,122,123]. Reduced plasma FGF19 concentrations are observed in obesity [124]. Because of these metabolic actions, FGF15/19 was suggested to be a promising therapeutic target for weight loss [125,126], but clinical proof of efficacy on body weight remains to be reported. FGF19 analogue has been used in clinical trials for diabetes and non-alcoholic steatohepatosis [127]. Further support of FGF15/19 utility in the clinic will require more in depth understanding of their differences in rodents and humans, especially given the controversial evidence concerning safety related to hepatocellular carcinoma (e.g., reviewed in Gadaleta, R. M [128]).

Interestingly, enterocytes in the small intestine produce a ghrelin receptor inverse agonist and antagonist, liver-expressed antimicrobial peptide 2 (LEAP2), first described in the liver [129]. It counteracts ghrelin-dependent orexigenic effects by dominant binding to the ghrelin receptor. Studies on mice showed that LEAP2 also reduces ghrelin secretion and hepatic glucose production, and its application results in reduced body weight and adiposity (reviewed in [130]). Secretion of LEAP2 depends on the metabolic state of the body being increased in obesity and reduced in fasting [131]. It has been suggested that LEAP2 is regulated in a manner diametrically opposite to that of ghrelin, and that increased levels of LEAP2 in the state of obesity contributes to ghrelin resistance [114]. One could hypothesize that LEAP2 has a protective effect against hyperphagia in a positive energy balance. Therefore, it would be interesting to investigate whether LEAP2 KO animals or patients with reduced LEAP2 plasma levels present more severe obesity. Hypothetically, it could be used to reduce the increase in plasma ghrelin levels seen after weight loss or bariatric surgery and thereby reduce the orexigenic effect of ghrelin to limit weight regain. However, LEAP2 inhibits growth hormone secretion [114] and it is unclear whether this can be a limiting factor in development of LEAP2-based therapies. Currently, the effect of LEAP2 on postprandial glucose metabolism and food intake is being investigated in humans (ClinicalTrials.gov: NTC04621409).

## 6. Intestinal Barrier Function

It is generally accepted that obesity is accompanied by low grade inflammation in the intestine [132]. The genesis of this condition is thought to be related to caloric overload. A study in humans shows that one month of high-fat feeding induces metabolic endotoxemia [133]. It is believed that high calorie foods sustain semi-pathogenic micro-organisms, and shift the balance away from the beneficial bacteria [134], mentioned above. Microbial pathogens can disrupt the tight junctions between the intestinal cells (Figure 4A) and thereby compromise the gut barrier function allowing the passage of endotoxins, which may induce systemic inflammation.

Moreover, studies in germ-free mice suggest that the microbiota promote local intestinal immunity through influencing the presence of intraepithelial lymphocytes (IEL) (Figure 4C).

IELs release cytokines that have detrimental effects on intestinal barrier function and may affect overall metabolism [135]. Certain polyunsaturated fatty acids, such as omega-6 and arachidonic acid (ARA), can serve as proinflammatory factors [136]; however, the pathological process is associated with the impaired metabolism of ARA in some individuals rather than with increased consumption of it [137]. In people living with obesity, impaired intestinal permeability and integrity (“gut leakiness”) does not necessarily manifest in clinical symptoms such as bleeding and malabsorption, which are often observed in other gastrointestinal conditions (e.g., Crohn’s disease, irritable colon disease or ulcerative colitis). Therefore, gut permeability is often overlooked as a potential contributor to metabolic disorders.

Systemic inflammation caused by gut endotoxins in high fat diet-induced obesity results in hypothalamic inflammation [138] and may drive increased food intake and consequently weight gain [139]. In addition, caloric overload and hypothalamic inflammation in obesity has been associated with dysregulation of lipolysis and lipogenesis in adipose tissue [140] and impairment of gluconeogenesis and glycolysis in the liver [141]. Specifically, in POMC neurons, inflammation causes a loss of glucose sensitivity, resulting in impaired control of glucose metabolism (reviewed in [142]). Thus, restoring intestinal permeability in obesity may reduce systemic inflammation and improve food intake control. One way to improve intestinal barrier function can be to prevent the infiltration of IEL in the intestinal mucosa. Studies on mice showed that a knockout of integrin β7 blocks integrin function for homing lymphocytes in the intestinal epithelium thereby preventing their migration to the intestinal border and reducing the inflammation. Interestingly, a study on mice showed that this intervention prevents weight gain on high fat diet and suggests that reducing inflammation can increase metabolic health in obese individuals [135].

Remarkably, some microbial products, for example, SCFAs, may also strengthen the mucosal barrier by improving the energy status of colonocytes, for which they are the main energy source [143]. Therefore, promoting SCFAs-producing microbiota can potentially also be used as therapeutic means for restoring intestinal integrity in obesity. As another example, clinical trial results suggest that methionine-restricted diets may help maintaining beneficial SCFA-producing microbial species resulting in increased fatty acid oxidation, reduced adiposity, but no significant weight loss [144].

Although systemic inflammation due to impaired intestinal barrier in obesity exacerbates the metabolic syndrome, it is not clear whether the prevention of bacterial endotoxin leakage into the circulation would induce weight loss or prevent weight gain [145]. Unfortunately, interventions directed at restoring gut permeability show some success for lower body weight maintenance in animal studies [146,147], but very limited efficacy in humans (reviewed in [148]).

## 7. Concluding Remarks

The gut epithelium may be an attractive therapeutical target for obesity treatment, since this tissue contains a large and regulatable pool of constantly self-renewing cells, controlling nutrient absorption and disposition, as well as food intake. Together with life-style interventions (which should always be considered as the first line of treatment), enhancement of satiety hormone secretion presumably holds the largest potential for pharmaceutical treatment of obesity. Other emerging strategies, such as microbiota tailoring or modulation of nutrient absorption beyond currently sold lipase-inhibitors, may also prove to be suitable for targeting obesity that and should be further explored.

Advanced in vitro platforms, such as organoids and organ-on-a-chip systems, will help further development of research programmes as an important asset for real-time studies on intestinal cell populations and for drug screening.

Targeting gut receptors has a clear advantage, as it allows for harnessing the gut-brain axis to reach the food intake centres in the brain without the actual drug passing the blood-brain barrier. Thus, the gut sensing machinery can be activated from the intestinal lumen (or from the blood stream) to amplify gut hormone secretion. This approach may be advantageous to injected gut hormone analogues, because secreted gut hormones will reach high local concentrations in the portal system and in the vicinity of the nerve endings of the gut-brain axis, which may amplify their satiety effect. The mechanisms of intestinal cell differentiation have recently been revealed, and we continue to get a better understanding of how to control the absorptive and secretory capacity of the intestine. This offers new opportunities for fine tuning and correcting the gut signalling to improve metabolic control. Thus, a pharmacological modulation of cell differentiation in the gut epithelium can increase the number of satiety hormone-producing cells [53,54,149]. New approaches, such as gene therapy and microbe-mediated drug delivery [150,151], are being developed to enable new therapeutic means in a safer way, and it will be exciting to see how these strategies will fare in the future.

Modulation of gut microbiota is now gaining popularity as personalized medicine in attempts for metabolic corrections. While we are still awaiting conclusive data from human studies as to what extent the alteration of microbiota may result in weight loss, the unique spectra of metabolites produced by gut microbiota may be useful to identify biomarkers for diagnosis and development of treatment strategies for metabolic disorders.

Tremendous progress has been made in recent years in understanding the details of cell renewal, regulation of cell protein synthesis and fine-tuning of cell function in the gut. We believe that exploiting intestinal signals has the potential to bring food intake into balance in a safe way and to improve metabolic health in individuals with obesity with long-lasting effects throughout life. Indeed, as mentioned throughout this review and as summarized in Table 1, several attempts has already been made to harness gut signalling/absorption for obesity treatment.

## Figures and Tables

**Figure 1 metabolites-12-00039-f001:**
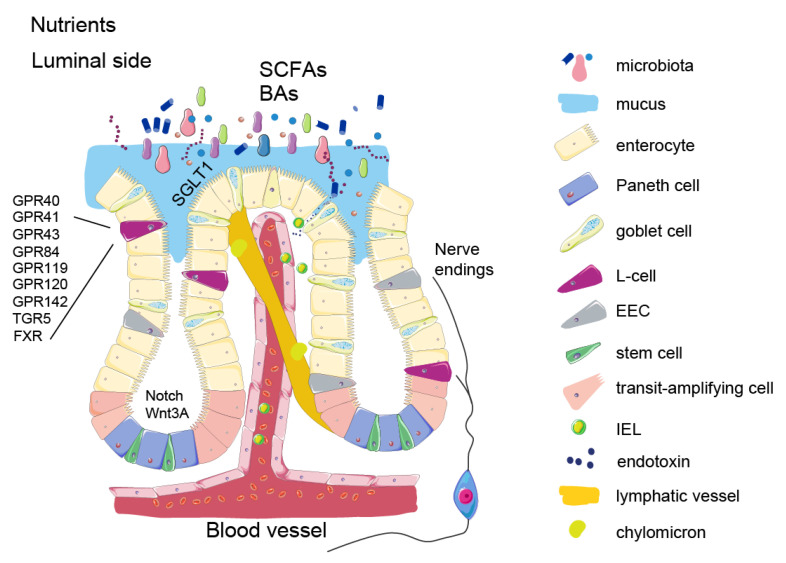
Schematic presentation of intestinal epithelium. Nutrients and metabolites entering the luminal side of the intestine are absorbed by enterocytes. Chylomicrons enters the lymphatic system through lacteals (described in *Signaling properties of nutrients* section). SCFAs and secondary bile acids are produced in the distal part of the small intestine and in the colon by various microbes (described in *Microbial metabolites* section). These molecules are sensed through specific receptors on EECs. In this figure, L-cells are presented separately from the other EECs to highlight their particular role in the food intake control and specific receptor expression (described in the *Enteroendocrine cells* section). Nerve endings originating from the nodose ganglion detect the gut hormone release and transmit the signal along the gut-brain axis to the appetite centers. The figure also shows impaired intestinal barrier function in obesity due to immune cell migration and endotoxin release (described in the *Intestinal barrier function* section).

**Figure 2 metabolites-12-00039-f002:**
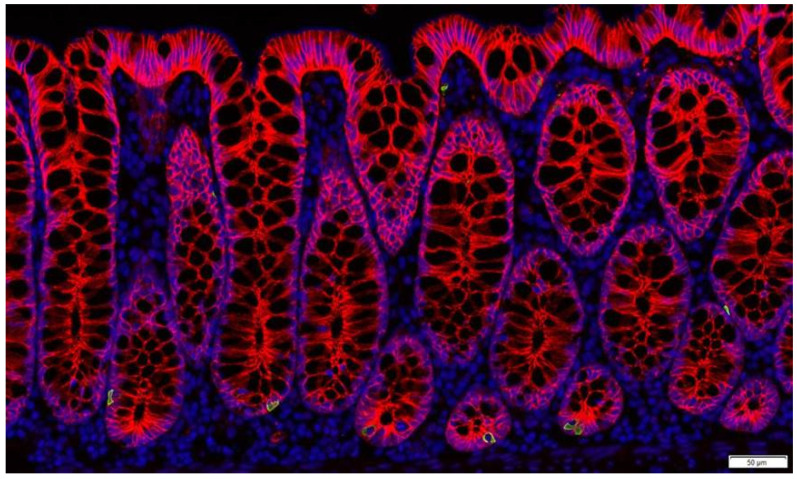
Enteroendocrine L-cells in non-human primate colon (Cynomolgus macaque). L-cells are identified by immunostaining for GLP-1 (green). Nuclei are labelled by DAPI (blue) and the intestinal cells are outlined by e-cadherin staining (red).

**Figure 3 metabolites-12-00039-f003:**
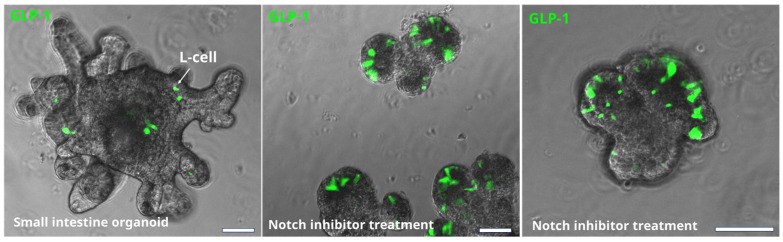
Effect of Notch inhibition on the number of EEC in mouse small intestine organoids. L-cells are identified by immunostaining for GLP-1 (green). Bars are 50 µm. Notch inhibition significantly increases the number of L-cell (and other secretory cells) in intestinal epithelium. However, it also has a negative effect on epithelial growth and crypt formation.

**Figure 4 metabolites-12-00039-f004:**
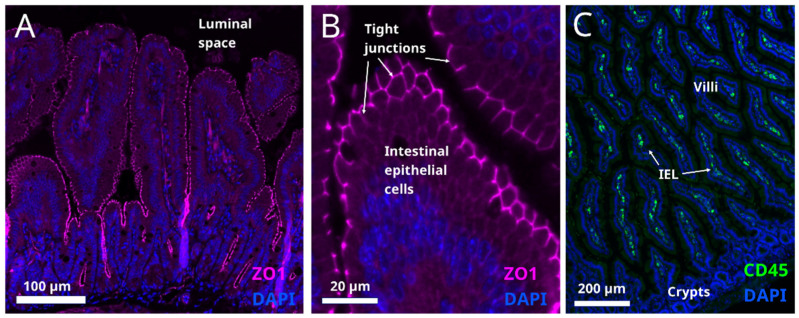
(**A**,**B**) Tight junctions are situated on the apical surface of the intestinal epithelial layer (cell nuclei are labelled with DAPI, blue) in mouse small intestine section. They are identified by immunostaining for ZO1, a tight junction marker, magenta labelling). Presence of microbial pathogens in the luminal space impairs the integrity of tight junctions (not shown). (**C**) Intraepithelial lymphocytes in mouse small intestine identified by immunostaining for their marker CD45 (green). Nuclear marker DAPI is shown in blue. In obesity, IELs infiltrate the intestinal epithelium and the cytokines released by IELs damage the cell connections in the epithelial layer (not shown).

**Table 1 metabolites-12-00039-t001:** Pharmacological compounds for obesity treatment mentioned in the review.

Target Pathway	Available Compounds	Biological Effect	Side Effects	References
SGLT1 inhibitor	Phloridzin, Sotagliflozin	Reduced glucose absorption	Osmotic diarrhoea	[26]
Lipase inhibitor	Orlistat	Reduced lipid absorption	Oily stool, diarrhoea	[25]
GPR119 agonists	AR231453 APD597 AS1669058	Increased production of satiety hormones	not determined	[41]
Goat Inhibitor	GLWL-01	Reduced ghrelin signaling	Mild nausea	[67]
TGR5 agonists	NT-777	Increased production of satiety hormones	Gall bladder enlargement	[77,78,79]
FXR agonist	Fexaramine	Increased FGF15/19 production, reduced inflammation, beneficial microbiota changes	Itchy skin (pruritis)	[80]
Probiotic dietary supplements	*Akkermansia muciniphila*, *Lactobacillus*, *Streptococcus*, *Bifidobacterium*	Increased gut hormone production, reduced inflammation	Gastrointestinal discomfort	[86]
FGF19	NGM282, Aldafermin (tested for NASH and type 2 diabetes treatment)	Suppression of hepatic lipogenesis, increase in adipose thermogenesis in the liver	Gastrointestinal discomfort	[127]
LEAP2	LEAP2 peptide	Anti-ghrelin action	Reduced growth hormone secretion?	Cl.Trials.gov Identifier: NCT04621409

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
