# Peer review of "Targeting the Gut in Obesity: Signals from the Inner Surface"

_metabolites, 2022, doi:10.3390/metabo12010039_

Round 1

Reviewer 1 Report

To the review was presented to me an review article about intestinal signaling in obese patients. The authors described all the important issues in a very thorough and detailed manner. The abstract of the work is written correctly - but it lacks the final conclusion. Moreover, the work does not have a typical aim for it that could be found at the end of the introduction. A large number of new scientific works were used in the work, which increases its cognitive value.

The main part of the work is also described in detail, but it is worth noting the introduction of interesting figures and tables that would summarize and diversify the main part.

In my opinion, chapter 4 should be supplemented with literature on specific probiotic strains important in obesity. In addition, each chapter should be followed by a summary of the authors and their reference to the topic.

The conclusions of the work require redrafting - they are a summary of the work, not an inference - they should be corrected.

Author Response

Reviewer 1.

  1. To the review was presented to me an review article about intestinal signaling in obese patients. The authors described all the important issues in a very thorough and detailed manner. The abstract of the work is written correctly - but it lacks the final conclusion. Moreover, the work does not have a typical aim for it that could be found at the end of the introduction. A large number of new scientific works were used in the work, which increases its cognitive value.

We thank Reviewer for overall positive feedback and for the opportunity to improve the manuscript. To clarify, we have mentioned both in the title and in the abstract that this review describes satiety signals from the intestinal epithelium, including current evidence of pathophysiological changes that may occur in these pathways in the state of obesity. We also address how these pathways are exploited in therapeutic treatment of obesity to control the food intake. As we do not believe that obesity results from impaired signalling from the gut, but rather from hedonic-driven overeating, we are focusing on the ways to harness the homeostatic mechanisms from the body’s natural feedback to nutrient intake. However, other changes in the gut resulting from obesity are likely to contribute to the wide array of complications associated with obesity, such as systemic inflammation and insulin resistance. We have now added these notes in the introduction and respective sections (p.2, EEC section on p.6). The changes in gut permeability and microbial content has been described in the original text. A conclusion was added to the abstract.

“Current therapies exploiting the appetite-regulating properties of the gut are based on chemically modified versions of the gut hormone, glucagon-like peptide-1 (GLP-1) or on inhibitors of the primary GLP-1 inactivating enzyme, dipeptidyl peptidase-4  (DPP-4). The effectiveness of these approaches shows that that the gut is a promising target for therapeutic interventions to achieve a significant weigh loss. Growing understanding of functionality of intestinal epithelium and new delivery systems will likely help developing selective and safe gut-based therapeutic strategies for obesity treatment in the future. Here, we provide an overview of the major homeostatic appetite-regulating signals generated by the intestinal epithelial cells and how these signals may be harnessed to treat obesity by pharmacological means.”

The aim of the review is now stated in the introduction:

“In this review, we will describe the satiety feedback signals generated by the intes-tine during digestion and how these pathways may be harnessed to control the food in-take and thereby produce weight loss.”

  1. The main part of the work is also described in detail, but it is worth noting the introduction of interesting figures and tables that would summarize and diversify the main part.

Thank you for this insightful point. We have added a new graphical abstract and three additional figures, visualising the points discussed in the text. Furthermore, we have added a table covering the therapeutic approaches described in the review.

  1. In my opinion, chapter 4 should be supplemented with literature on specific probiotic strains important in obesity. In addition, each chapter should be followed by a summary of the authors and their reference to the topic.

Point well taken. We have now added a paragraph summarising the use of main probiotic strains and added conclusions to each section. In addition, we also added examples of microbiota changes associated with obesity and after gastric bypass. We felt that going into the depth of individual bacterial strain qualities would be beyond the purpose of this review.

  1. The conclusions of the work require redrafting - they are a summary of the work, not an inference - they should be corrected.

We have now rewritten the conclusion section and incorporated our view on the perspectives and advantages of targeting the intestine as anti-obesity therapy.

The changes to the manuscript are quite substantial, thus we do not indicate all the added lines in this letter, but made a mark-up version with new text in blue.

Reviewer 2 Report

The first impression reading the article title is "okay, definitely a paper focusing on the role of the gut in obesity!"... but reading the article, these expectations are lost. Authors include in all sections only well-known information on anatomy and physiology, but no innovative remarks are provided.

In this review report, I don't want to reject this article since I believe the focus treated by Authors is very interesting and appealing, but I think it should be better discussed. For example, possible treatment approaches should be proposed and discussed... but not only pharmacological approaches but also nutritional strategies, since the first approach for the treatment of obesity is the diet (and this should never be forgotten).

For these reasons my final decision is "Reconsider after major revision", with the hope that the Authors reformulate the entire paper.

In addition, as very minor concerns:

  • at line#30: the table of content is not necessary, since this is a journal article and not a book chapter
  • the introduction section provides well-known concepts of anatomy. This is all so superfluous and redundant. It should be replaced by introductory concepts on the focus of the review paper

Author Response

Reviewer 2.

The first impression reading the article title is "okay, definitely a paper focusing on the role of the gut in obesity!"... but reading the article, these expectations are lost. Authors include in all sections only well-known information on anatomy and physiology, but no innovative remarks are provided.

We thank the Reviewer for evaluating the review and for the insight full and constructive points raised. We feel we need to clarify the purpose and the content of the review.

From the title of our work, we are describing the gut satiety signals and approaches to harness these intestine-derived signals for correction of food intake in obesity. As we do not believe that intestine-derived signals have a major role in driving obesity, we describe data on structural and functional changes in the intestine during obesity, as those are likely contributing to obesity related co-morbidities such as systemic inflammation and insulin resistance.

With regards to anatomy and physiology details, we felt that mentioning the structural and physiological characteristics of this organ would help the reader to better understand the mode of action of these therapeutic approaches. Importantly, our review provides a thorough description of recent advances in the field and includes clinical trial information on efficacy of these approaches. In agreement with Reviewer’s comment, we have expanded the “innovation” part and reduced “anatomy and physiology” descriptions throughout the review (text in blue in the markup version).

In this review report, I don't want to reject this article since I believe the focus treated by Authors is very interesting and appealing, but I think it should be better discussed. For example, possible treatment approaches should be proposed and discussed... but not only pharmacological approaches but also nutritional strategies, since the first approach for the treatment of obesity is the diet (and this should never be forgotten).

We have now made additions according to reviewer’s wishes, also describing nutritional strategies, which target the satiety input and overall energy intake (lines 218-223; 140-146; 62-62). Initially, our plan was to focus on the pharmacological treatment strategies only, but the additions proposed by the reviewer were interesting and inspiring. We specifically did not focus on the calorie-restricting approaches this time, as we felt this would not be targeting the satiety mechanisms of the gut (in fact the contrary) and therefore off-topic.

New potential approaches are mentioned in the original text (199-202 580-583) and new text added in lines 213-217, 319-341, 453-458, 532-536 and in Concluding remarks.

For these reasons my final decision is "Reconsider after major revision", with the hope that the Authors reformulate the entire paper.

We have now removed a large part of the anatomical details and added new paragraphs throughout the paper (text in blue in the manuscript). As the changes are quite substantial, we do not indicate all the added lines in this letter. We added 3 new figures and a table to better illustrate the basis of the therapeutic approaches.

In addition, as very minor concerns:

  • at line#30: the table of content is not necessary, since this is a journal article and not a book chapter

The table of content is now removed.

  • the introduction section provides well-known concepts of anatomy. This is all so superfluous and redundant. It should be replaced by introductory concepts on the focus of the review paper

 The intro is now re-written according to the reviewer’s recommendations. We hope that you find the changes to the manuscript appropriate and wishing you a happy New Year.

Round 2

Reviewer 1 Report

The authors' responses and significant changes to the manuscript are sufficient to accept the current version of the article - it makes it possible to publish it.

Reviewer 2 Report

The authors addressed all the issues raised. The article is now more readable and better organised.

In my opinion, the article is now suitable for publication.

Only one concern: are the figures used free of copyright permissions?